# Characterization of Rock Samples by a High-Resolution Multi-Technique Non-Invasive Approach

**Silvana Fais** [1,2], **Francesco Cuccuru** [1], **Giuseppe Casula** [3,*], **Maria Giovanna Bianchi** [3] and **Paola Ligas** [1]

1   Department of Civil and Environmental Engineering and Architecture (DICAAR), University of Cagliari, 09123 Cagliari, Italy; sfais@unica.it (S.F.); cuccuruf@unica.it (F.C.); pligas@unica.it (P.L.)
2   Consorzio Interuniversitario per l'Ingegneria delle Georisorse (CINIGeo), 00186 Roma, Italy
3   INGV Istituto Nazionale di Geofisica e Vulcanologia, Sezione di Bologna-Via Donato Creti, 12-40128 Bologna, Italy; mariagiovanna.bianchi@ingv.it
*   Correspondence: giuseppe.casula@ingv.it; Tel.: +39-51-4151-415

**Abstract:** Three different non-invasive techniques, namely Structure from Motion (SfM) photogrammetry, Terrestrial Laser Scanner (TLS) and ultrasonic tomography integrated with petrographic data, were applied to characterize two rock samples of a different nature: A pyroclastic rock and a carbonate rock. We started a computation of high-resolution 3D models of the two samples using the TLS technique supported by a digital SfM photogrammetry survey. The resulting radiometric information available, such as reflectivity maps, SfM photogrammetry textured models and patterns of geometrical residuals, were interpreted in order to detect and underline surface materials anomalies by a comparison of reflectance and natural colour anomalies. Starting from the 3D models from previous techniques, a 3D ultrasonic tomography on each rock sample was accurately planned and carried out in order to detect internal defects or sample heterogeneity. The integration of the above three geophysical non-invasive techniques with petrographical data—especially with the textural characteristics of such materials—represents a powerful method for the definition of the heterogeneity of the rocks at a different scale and for calibrating in situ measurements.

**Keywords:** non-invasive techniques; comenditic pyroclastic rock; Pietra Forte carbonatic rock; terrestrial laser scanner; photogrammetry; 3D acoustic tomography; petrographic data; integrated interpretation

## 1. Introduction

Establishing the properties and state of conservation of stone materials can relate to many fields of research, from cultural heritage to architecture and also geological studies [1–13].

To study stone materials both outcropping and in depth, with appropriately prepared samples one can make as many measurements as necessary with different techniques. Moreover, some characterization analyses are destructive and there is a limit to the number of samples that can be sacrificed. In the analysis of rock samples, non-destructive techniques are constantly being improved and new ones have been introduced. Rock sample analysis is becoming more widespread with the introduction of sophisticated methods of analysis such as X-ray computed tomography (X-ray CT) and μ-XRF [14–17]. These high-tech methods provide a huge quantity of micro structural and microscopic information. Zhang et al. [18] have used X-ray CT to obtain 3D images of shale samples and starting from 3D X-ray CT images they used the Finite Element Method (FEM) to compute the elastic properties under dry, water-saturated, and oil saturated conditions. However, the above

recent approaches require expensive equipment that cannot always be available at a given laboratory. Some techniques such as the Terrestrial Laser Scanner (TLS) [19] and fluorescence LIDAR [20] are very useful to analyse the shallow parts of stone materials. Other non-invasive geophysical techniques such as magnetic resonance imaging (MRI) [21] and infrared thermography (IRT) are useful to probe just below the surface materials [22,23] while ultrasonic techniques penetrate inside the materials [24–27]. Variations in ultrasonic wave propagation velocity and amplitude are related to rock internal structure changes [28]. In the non-invasive characterization of stone materials, the integration of different types of complementary information can improve the analysis and the diagnostic process [29]. This is of paramount importance in many research fields such as heritage science, engineering geology and mining. However, it should be pointed out that efficiency in the multi-technique and multiscale integrated approaches is possible if and only if the integration is performed appropriately and combined with a good knowledge of the petrographic characteristics of the stone materials.

In this work a high-resolution multi-technique non-invasive approach based on the integration of the ultrasonic technique and high-resolution 3D models is proposed for a detailed geophysical characterization of rock samples. The improvement of surveying solutions to obtain high-resolution 3D models of the investigated objects gives an adequate geometric support in rendering the 3D ultrasonic data at their precise location. In this way it is possible to increase the accuracy of the dynamic elastic characterization of the rock samples while replacing some of the ordinary destructive tests. A similar integrated approach was recently applied in the cultural heritage field [11,30], but it has never been applied in the geophysical characterization of rock samples.

The samples to be investigated need to be carefully selected to ensure they are representative of the rock types under study. In this study, two rock samples of a different nature were analysed: Namely a pyroclastic rock (comendite) and a shelf limestone (Pietra Forte).

In order to obtain natural color texturized 3D models for the characterization of the investigated rock samples, we started a computation of high-resolution 3D models using the Terrestrial Laser Scanning (TLS) technique integrated by a Structure from Motion (SfM) photogrammetry survey.

TLS technology is based on the repeated measure of distances between the emitter of a laser ranging system and the reflecting surface of the illuminated objects. Since TLS is a highly automated system, it can produce several point clouds of millions of points per second of objects of a variable scale and complex shape all around the field of view [31].

Structure from Motion (SfM) is a technology based on computer vision by which the photogrammetric reconstruction from sets of images is possible. In particular, high-resolution 3D models with millions of points can be computed from overlapping images using, for example, calibrated digital cameras [32,33].

High-resolution 3D models computed with integration of TLS and SfM techniques provide precise information about the geometrical shape of studied objects, but also on a radiometric parameter called reflectivity or reflectance, a dimensionless ratio between the energy backscattered by the surface of the illuminated bodies and sensed by laser diode phase comparator and the incident energy emitted by the laser itself. The reflectivity varies with the colour of bodies but is also proportional to the roughness of the illuminated surfaces, and is often used to classify different component materials [9,11,19,34–38].

In the first step of our procedure the radiometric information on the samples were obtained: Namely reflectivity maps, SfM photogrammetry textured models and patterns of geometrical residuals. These data were interpreted by a comparison of reflectance and natural colour anomalies with the conditions of the surface materials in order to locate and underline possible anomalies due to defects, oxidation, small cavities, fractures, alteration and differences in texture and composition. Compared with the macroscopy analysis which strictly depends on the personal expertise of the analyst, the above techniques guarantee scientific reproducibility and objectivity.

In the second step of this workflow, based on the 3D models from previous techniques, a 3D ultrasonic tomography on each rock sample was accurately planned and carried out. The use of information based on the propagation of ultrasonic signals through the investigated materials represents

one of the most common approaches across all scales for the elastic-mechanical characterization of rocks [26,39–44]. The ultrasonic methods have a wide spectrum of applications both in situ and in laboratory producing higher resolution results [45–48].

The 3D ultrasonic tomography leads to the fine characterization of the rock samples in terms of elastic properties and helps to detect defects and/or heterogeneities inside them. Velocity variations can be related with variations in the petrographical characteristics (especially texture) and/or with zones of different elastic-mechanical conditions. In fact, the presence of voids, fissures, discontinuities and alterations has a significant effect in slowing the propagation of the ultrasonic signal by absorbing its energy.

Furthermore, significant benefits in geophysical stone characterization can be obtained from the results by the above three non-invasive techniques integrated with petrophysical and petrographical data, especially composition and texture. These results can be related to many features of interest, depending on the target of the study.

## 2. Materials and Methods

### 2.1. Sample Materials

In order to calibrate and optimize the interpretation of the data from the high-resolution multi-technique approach proposed in this work, the investigated materials were also analysed from the petrographical point of view and especially for their textural characteristics.

Textural characteristics greatly influence the physical behaviour of the rocks [49–57] and the response of the geophysical non-invasive methods. In this study, two very different lithotypes of a pyroclastic and a carbonate nature were investigated by the above techniques. A pyroclastic rock, classified as comendite [58,59] from the Sulcis area (south-western Sardinia, Italy) and a shelf limestone known as Pietra Forte, belonging to the "Calcari di Cagliari Auct" [60,61], outcropping in the hills of the town of Cagliari (south Sardinia, Italy) were analysed.

The geophysical measurements were performed on prismatic samples of the dimensions of 10.0 cm × 9.8 cm × 22.0 cm for comendite and the dimensions of 10.4 cm × 12.1 cm × 24.5 cm for Pietra Forte [62,63]. A part of each was used to prepare both the thin sections for optical microscopy (OM) and scanning electron microscopy (SEM) analyses, and a small cubic sample (1.5 cm in size) for mercury porosimetry injection.

The comendites are one of the products of the Sardinian Oligo-Miocene calkalcaline volcanic cycle associated with the opening of the Balearic basin and the drift of the Sardinian–Corsican Block [64–66], while the Pietra Forte represents one of the sedimentary products deposited in the Fossa Sarda Auct. during the transgressive phase (third sedimentary cycle) of the Upper Miocene [67].

The comendites are rocks with a good workability, often used as ashlars in the local historical construction sector. This material can be found in alternation with other volcanic products in the heterogeneous walls of the historical-architectural buildings of the villages in Sulcis.

The Sulcis comendites are generally peralkaline rhyolites [68] macroscopically made up of welded pyroclastic flows, of both flow and fall poorly welded deposits, characterized by the presence of sanidine and quartz phenocrysts especially in welded deposits, and of small lithic elements and lapilli of millimetric–centimetric dimensions in poorly welded deposits. Due to the presence of glass and high values of porosity, comendites can be affected by severe forms of exfoliation, alveolation and fracturing, especially in the poorly welded facies.

The Pietra Forte is a very consistent and tenacious carbonate rock, widely used in the past to build monumental structures and the ancient city walls of the town of Cagliari [26,69].

The Pietra Forte is a well litified shelf limestone of a biohermale–biostromal nature characterized by a very abundant fossiliferous content with the prevalence of Lithothamnium algae and organogenic remains of bivalves, gastropods, echinoids, bryozoans, crustaceans and fish [70]. According to the

classification of the carbonate rocks proposed by Folk (1959) [71], the Pietra Forte can be classified as a biolithite, while according to Dunham (1962) [72] it is a boundstone.

The depositional environment of Pietra Forte is characterized by coastal and infralittoral conditions with high energy and paleobathymetry lower than 30 m [73].

At the outcrop scale the Pietra Forte is fractured and locally affected by karst phenomena that develop in the subsoil of Cagliari giving rise to an aquifer characterized by a turbulent water regime. When used as a building material, the Pietra Forte can be subject to a degradation that generally occurs with fracturing and solubilization processes that cause an increase in pore size.

*2.2. Methods*

2.2.1. Petrographic Analysis

Optical and scanning electron microscopy were used for the analysis of the compositional, textural and some petrophysical characteristics of the studied rocks, such as porosity. Optical microscopy (OM) was performed with a Carl Zeiss Axioplan microscope (Carl Zeiss, Oberkochen, Germany). The thin sections studied in OM were treated with blue dye epoxy resin in order to improve the identification of the porous system [74].

Scanning electron microscopy (SEM) analyses was performed with a Zeiss EVO 50 VP model (Carl Zeiss, Oberkochen, Germany) connected to an EDS X-Max (Oxford Analytical Instruments Ltd., High Wycombe, UK). For the SEM analyses the samples were metal coated creating a conductive 10 nm layer of gold on their surface to inhibit charging, reduce thermal damage and improve the secondary electron signal required for morphology examination.

2.2.2. Mercury Porosimetry Analysis

The studied samples were also analysed by mercury intrusion porosimetry (MIP). This technique provides information on the effective porosity, the pores-throat diameters/radii and other pore structural parameters such as permeability and tortuosity. Additionally, bulk and skeletal densities were obtained by MIP analysis. As is well known, bulk density is defined as the ratio between the mass of the dry bulk sample and its bulk volume, comprising volumes of solid particles, open and closed pores. The skeletal density is defined as the ratio between the mass of the dry bulk sample and its apparent volume, that is the total sample volume. The MIP technique is based on the physical principle that the mercury—a non-reactive, non-wetting liquid—will not penetrate into fine pores until an adequate pressure is applied to force its entry into the analysed material. The pressure required for penetration of mercury inside the pores is inversely proportional to their size. The well-known relationship between the applied pressure and the pore-throat radius/diameter into which mercury is intruded is the Washburn equation [75]. For the MIP analyses, samples of cubic shape with the largest dimension of about 1.5 cm were prepared. Micromeritics Autopore IV 9500 (Micromeritics Instrument Corporation, Norcross, GA, USA) was used to measure the accessible porosity for pore-throat diameters ranging from about 3 nm to 360 μm.

2.2.3. Terrestrial Laser Scanner and SfM Photogrammetry

As previously stated the integrated application of TLS and SfM Photogrammetry is usually performed to compute high-resolution 3D models of studied complex shape bodies. As a matter of fact, 3D models are a useful facility to preserve the memory of geometrical shapes under study and for purposes of cultural heritage conservation. In fact, such models represent a valuable tool that can be exported easily, 3D printed and successively inspected by experts and scientists of various disciplines. Portable document format (PDF) dynamic 3D representations and movies of samples but also of complex shape bodies can be made and represent useful tools for operators and technicians that need to study complex shape bodies on their own office computers without any direct contacts with the bodies themselves, see for example [76] ([37,77,78] and references therein).

TLS technology, for example, measures distances between the emitter of the laser ranging system and the reflecting surface of the illuminated objects to produce point clouds over the target field of view. The phase-based systems are generally the most used in architecture due to their high accuracy and capture speeds [31].

In our study, a TLS survey was scheduled and executed to obtain some high-resolution point clouds to detect information about the precise point positioning and the reflectivity or reflectance, which as mentioned previously, represents the fraction of the total radiating flux incident upon a surface that is reflected and that varies according to the wavelength distribution of the incident radiation. In particular, in our case the reflectance is a dimensionless parameter detected as the ratio between the energy reflected by the surface of the target object and the energy emitted by the laser beam and varies in the range 0–1 [19,34–36,38]. Following the above consideration, the TLS survey was made with a Leica HDS-6200 scanner (Figure 1a) which has the option Higher scan density, which increases the range at which smaller objects and targets can be accurately modelled. The TLS survey was supported by Structure from Motion (SfM) photogrammetry in order to obtain a corresponding digital metrically correct colour image of the samples.

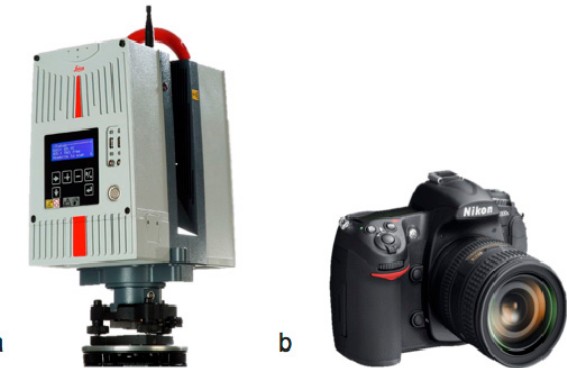

**Figure 1.** (**a**) The Leica HDS-6200 TLS. (**b**) The digital reflex camera Nikon D300.

As is well known, SfM photogrammetry is a passive technique. This means that the results are highly influenced by the quality of acquired images, sensors, acquisition schemes and settings of the cameras. For this reason, about 30 to 50 photos were taken carefully by a digital single lens reflex (SLR) camera (model NIKON D300 with sensor CMOS 23.6 mm × 15.8 mm, 12.3 MegaPixel) (Figure 1b) to get an almost complete 360° sequence of pictures, and particular attention has been taken to maximise overlap between successive photography positions adopting short camera baselines (i.e., distances between poses). Finally, a high-resolution natural colour 3D model was built with PhotoScan (Agisoft 2018©) using SfM photogrammetry and Multi View Stereo (MVS) matching or similar methodologies [79]. In particular, a three-step procedure was applied: First a few key-points were identified on the scenes common to all photographs, and then used to build a low-density point cloud as input to the densification process of the final step. The iterative procedure of key-points identification was based on a sparse bundle adjustment or similar algorithm that uses a non-linear least-square method to identify camera positions and then generate the low-density point cloud [32,80]. In the third final step smooth and height-field methods based on pair-wise depth map computation and MVS matching procedures were applied. The noisy data were filtered out and the useful pixel of all images were used to interpolate the sparse point cloud in order to generate a dense cloud texturized with natural RGB colours of the studied objects. As is well known, the resulting SfM 3D dense point cloud lacks scale and adjusted coordinates; Therefore, a seven parameter Helmert transform (rotation, translation and scale) has to be applied, using as reference the 3D model computed with TLS by means of suitable ground control points (CGPs), i.e., corresponding homologues points inside the prospect of the analysed samples [31,33,80].

Conversely, the processing of the TLS data can be summarized as follows: Point clouds input and format conversion; point clouds automatic filtering or manual editing (with the elimination of data out of tolerance and unusable points); Cloud to cloud draft alignment and fine registration with Iterative Closest Point (ICP) algorithm cloud aggregation [81]; 3D modelling and computation of morphological maps. In Figure 2 the flow chart of the previously described integrated procedures is reported. The final high-resolution 3D colour models (Figure 3a,b) from SfM photogrammetry complemented by TLS reproduce a faithful scale copy of the analysed samples and can therefore be associated with the surface geometric anomalies and the reflectivity effectively.

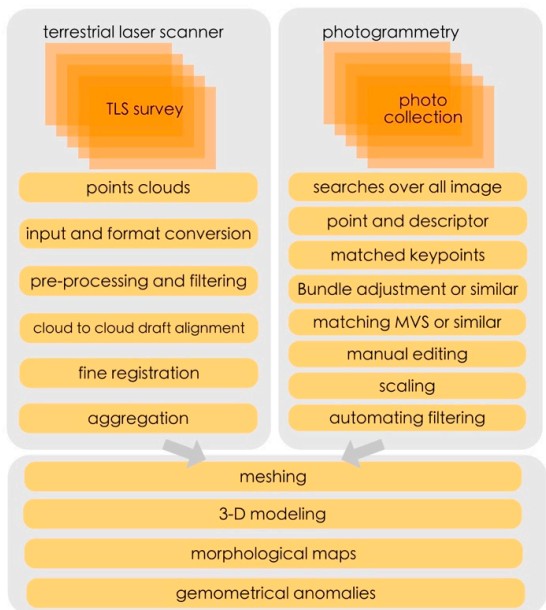

**Figure 2.** Flow chart of TLS and photogrammetry data acquisition and processing.

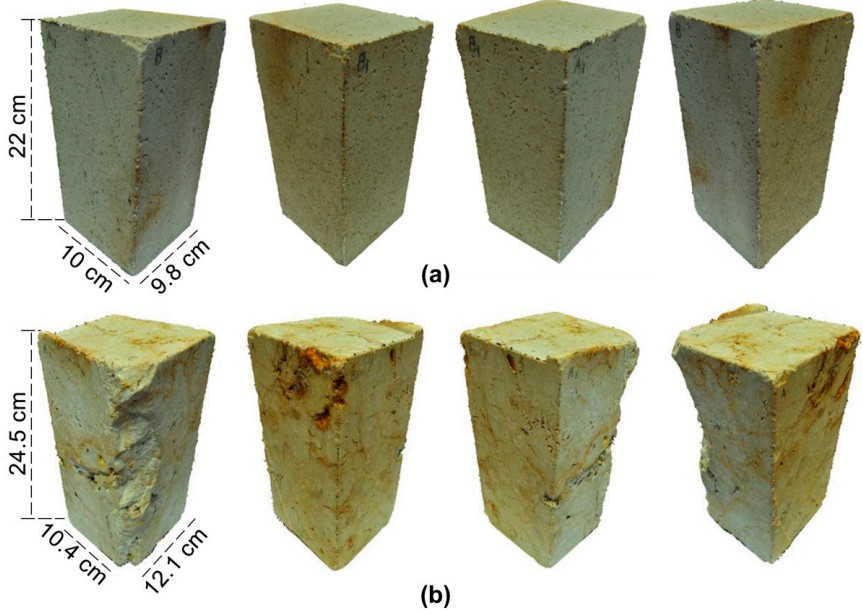

**Figure 3.** Different viewpoints of the samples reconstructed with the photogrammetric technique; (**a**) Comendite pyroclastic rock. (**b**) Pietra Forte carbonate rock.

In particular, during the processing stage of TLS and SfM photogrammetry data, the highest density options were used to compute the highest resolution 3D models with the aid of powerful

graphic processing units (GPU) and multicore central processing units (CPU) styled workstations. The resulting aggregated, filtered, registered, and unstructured point clouds consist of 10 to 20 million points for every sample analysed, and the computed meshes of clouds and graphic primitives were created with the smallest possible steps. As a matter of fact, the software programmes used for cloud editing and processing are integrated with the capability to perform in parallel all processes between available CPUs, thus reducing computing time.

Geometrical anomalies, were computed with the aid of the inspection facility of the software JRC 3D Reconstructor® property of Gexcel® using the following procedure: The area of the aggregated point cloud to be inspected was selected and saved, then a graphic primitive was adjusted in the least-square sense to the selected part of the point cloud to be inspected. In general, the choice of the graphic primitive is done in order to use a geometrical shape as similar as possible to the prospect to be analysed; plane or set of planes is used for walls or objects with faceted surfaces (walls, rectangular section pillars or parallelepiped shaped bodies).

Conversely, cylinders are often used to fit cylindrically shaped bodies such as columns; while spheres are used for spherical or semi-spherical bodies (domes and apses). In the latest version of the inspection facility conical geometrical primitives are also available.

In a second stage, the reference graphic primitive is meshed with the high-density modality using Delaunay triangulation or similar algorithms, and for every point of the inspected cloud element the distance from the reference model is measured. In fact, the difference (residual) between the point position of the inspected point cloud and the position of the reference geometrical primitive (i.e., a plane in our case) is computed in an arbitrary reference frame adopting the centre of the aggregated point cloud as origin. In our case, the resolution of the method is well below mm (see for example [77] and references therein) and the above described inspection procedure was applied to compute the geometrical anomalies of the aggregated point clouds (expressed in meters) as residuals with respect to a best fitted plane adopted as reference (see Figure 4). The geometrical anomalies computed in our case vary in the range ±2 mm. Finally, the radiometric parameter of reflectivity was used as an alternative to natural colours to texturize the sample maps shown in Figure 4. Since this parameter is a percentage, it varies between 0 and 1. The reflectivity or reflectance is a typical colourisation of the laser scanner styled point clouds but it can also be estimated effectively from the natural colorized 3D models detected with SfM photogrammetry (See for example [32,33,80]), in particular with the aid of the latest version of the Reconstructor® software (i.e., Reconstructor® versions 3 and 4 [82,83]).

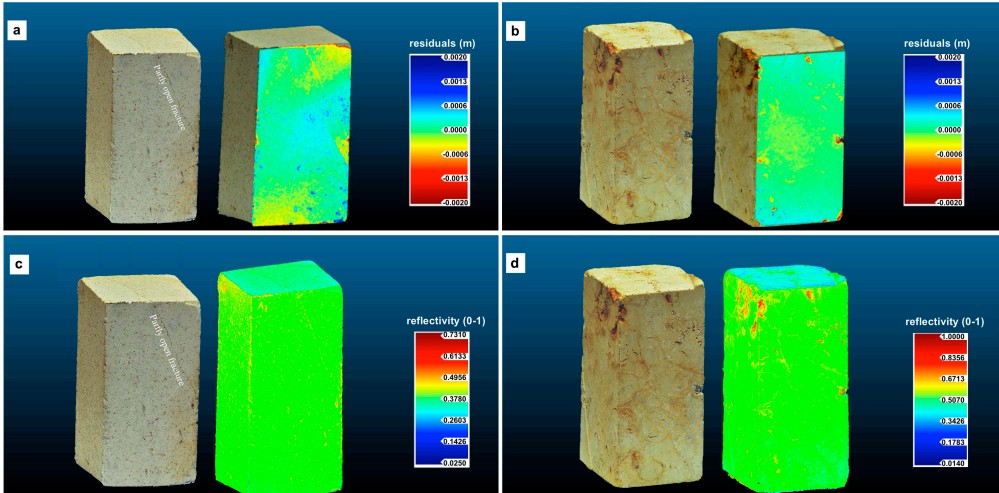

**Figure 4.** Results of overlapping of the photogrammetric survey with the TLS survey, elaborated first with the JRC 3D Reconstructor® software and then with Cloud Compare. Left: The Comendite sample. (**a**) geometrical anomalies (m) on a face of the sample. (**c**) reflectivity map. Right: The Pietra Forte sample. (**b**) geometrical anomalies (m) on a face of the sample. (**d**) reflectivity map.

### 2.2.4. 3D Acoustic Tomography

Based on the 3D models with previous techniques (TLS and SfM photogrammetry), a 3D ultrasonic tomography was planned designing an optimal survey and providing a very good spatial coverage of the investigated samples. The 3D tomography was carried out for a fine characterization of the samples in terms of elastic properties and in order to determine size and location of potential defects, cracks, fissures and cavities in their inner part. As a matter of fact, as recognized in previous works [8,26,27,45,84–87] rock texture, different type of porosity, compositional heterogeneities, and presence of fractures play an important role in ultrasonic measurements and greatly influence the elastic behaviour of the rocks. The great volume of ultrasonic data from the 3D tomography allows a much better understanding of the three-dimensionality of possible defects, fractures and heterogeneities inside the samples.

Ultrasonic measurements were carried out by the transmission method according to the ISRM 2007, 2014 [62,63] using a portable Ultrasonic Non-Destructive Digital Indicating Tester (PUNDIT Lab plus) device (Proceq, Schwerzenbach, Switzerland) interfaced to an oscilloscope (Fluke 96B) for the acquisition of the digital signals to be displayed and processed (Figure 5).

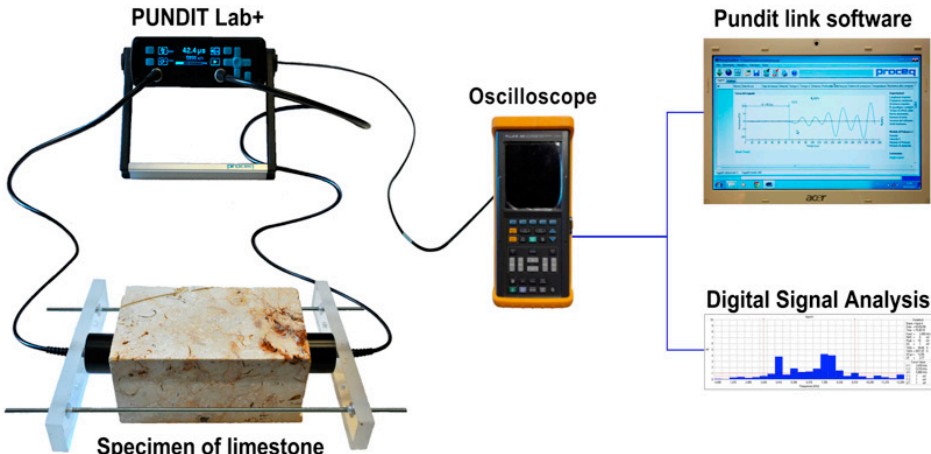

**Figure 5.** Setup used for the ultrasonic measurements.

The transducers were piezoelectric with a frequency of 82 kHz. The silicone snug sheets were used as the best coupling agent [88] to guarantee a good contact between transducers and materials. The transducers were placed on all sides of each sample with 20 mm of vertical spacing along parallel vertical profiles in such a way as to surround the entire sample. The transit time of the longitudinal ultrasonic waves between stations on the opposite faces of the sample was measured along a great number of measurement paths (around 2900) and averaged over five determinations. The mean of the measurements was computed not considering values with a deviation higher than 4%. Each station was alternatively used as a transmitter and a receiver located on the opposite faces of the samples [62,63]. The travel time of the longitudinal ultrasonic signals along the ray paths (source-receiver paths) was recorded together with the relative position and orientation of each ray in the investigated sample. The ultrasonic travel time was measured with an accuracy of 0.1 μs. The longitudinal wave velocity is the ratio of the travel distance to transit time of the longitudinal ultrasonic signal through the rock sample.

The ultrasonic travel time data volume was inverted exploiting the simultaneous iterative reconstruction technique (SIRT) [89–91] to obtain a 3D representation of the distribution of the longitudinal wave velocity inside the investigated samples. The iterative reconstruction technique indicates a series of successive approximations to correct an arbitrary initial parameter distribution (starting velocity model). In order to obtain a realistic and not arbitrary starting velocity model as input for the SIRT to invert travel times, we used a methodology based on the cross-correlation function (CCF) and described in a previous work [92]. Therefore, the cross-correlation function was computed

by a computer code developed in the laboratory of the Solid Earth Geophysics and Diagnostics at University of Cagliari (Italy) and used as a constraint to the SIRT tomographic inversion to include prior knowledge of the investigated sections. In fact, the SIRT algorithm starts calculation from an initial velocity model and follows with the computation of the travel times for each source/receiver path. After this, it compares the calculated and picked times. A correction factor for the time discrepancy between these times (calculated and picked) is applied to the velocity value of every cell (voxels) affecting the models. In this way the initial velocity model is modified by repeated cycles and the iterative process goes on until the desired accuracy is achieved. In synthesis the procedure is essentially made up of the following steps: Forward computation of model travel times, calculation of residuals and application of velocity corrections to the 3D volume of the voxels within the model. The 3D rendering of the resulting velocity distribution inside the samples was made using the Voxler software vers. 4.3.771 by Golden Software.

## 3. Results and Discussion

### 3.1. Thin Section Analysis and Mercury Porosimetry

The sample of comendite has characteristics typical of a pyroclastic rock with a hypocrystalline porphyritic texture mainly for sanidine and quartz phenocrysts (Figure 6a) with dimensions between 150 μm and 3 mm. The groundmass is characterized by an eutaxitic texture with presence of devitrification (Figure 6b), spherulitic structures and shards (Figure 6c). The visible porosity (25%), mostly related to the fluidal vitreous groundmass, is mainly characterized by channels with an aperture of 5 μm on average (Figure 6d,f).

The Pietra Forte is a very compact and poorly porous (2%) shelf limestone rich in Lithotamnium algae and well cemented by sparry calcite (Figure 7a). The visible porosity observed at the meso-microscale is mainly of secondary type and is characterized by fractures with apertures of 30 μm on average (Figure 7b) or it is due to dissolution phenomena that caused the development of moldic and vug porosities [93]. The sizes of the moldic pores, often centimetric, depend on the dimension of the dissolved bioclasts, while the vug dimension, usually millimetric, is related to the degree of dissolution (Figure 7c).

As observed from SEM analyses, sometimes the carbonate cement can be affected by an intercrystal porosity (Figure 7d,e) due to an incomplete cementation during or after the diagenesis. These pores have size of 2 μm on average.

In addition to the prevalent secondary porosity a primary porosity of the growth framework type [93] is present and is related to the algal growth structures (Figure 7f). The primary pores, having a dimension of 20 μm on average, are enclosed within the algal structures and are scarcely interconnected with the secondary pores.

The results of the MIP analyses are reported in Figure 8a,b and some of the most important parameters on the pore structures of the studied rocks obtained by MIP are reported in Table 1. The MIP curve in Figure 8a shows the pore-throat size distribution for the comendite sample. As can be observed, pore-throat diameters mainly range between 1.50 μm and 10 μm. The median pore-throat diameter turns out to be 2.09 μm (Table 1). Bulk and skeletal densities are 1.74 g/cm$^3$ and 2.30 g/cm$^3$, respectively (Table 1). The effective porosity value of 24.61% (Table 1) is high and in agreement with the OM and SEM analyses. Despite the high values of pore-throat sizes and porosity, the permeability value (1.53 mD) is low (Table 1). In this regard it should be pointed out that the tortuosity value (36.22, Table 1) indicates the articulated geometry of the porous system, which tends to hinder the circulation of fluids within the rock. This value is compatible with the information deduced from SEM analyses which showed a very articulated network of channels in the fluidal vitreous groundmass (Figure 6d–f).

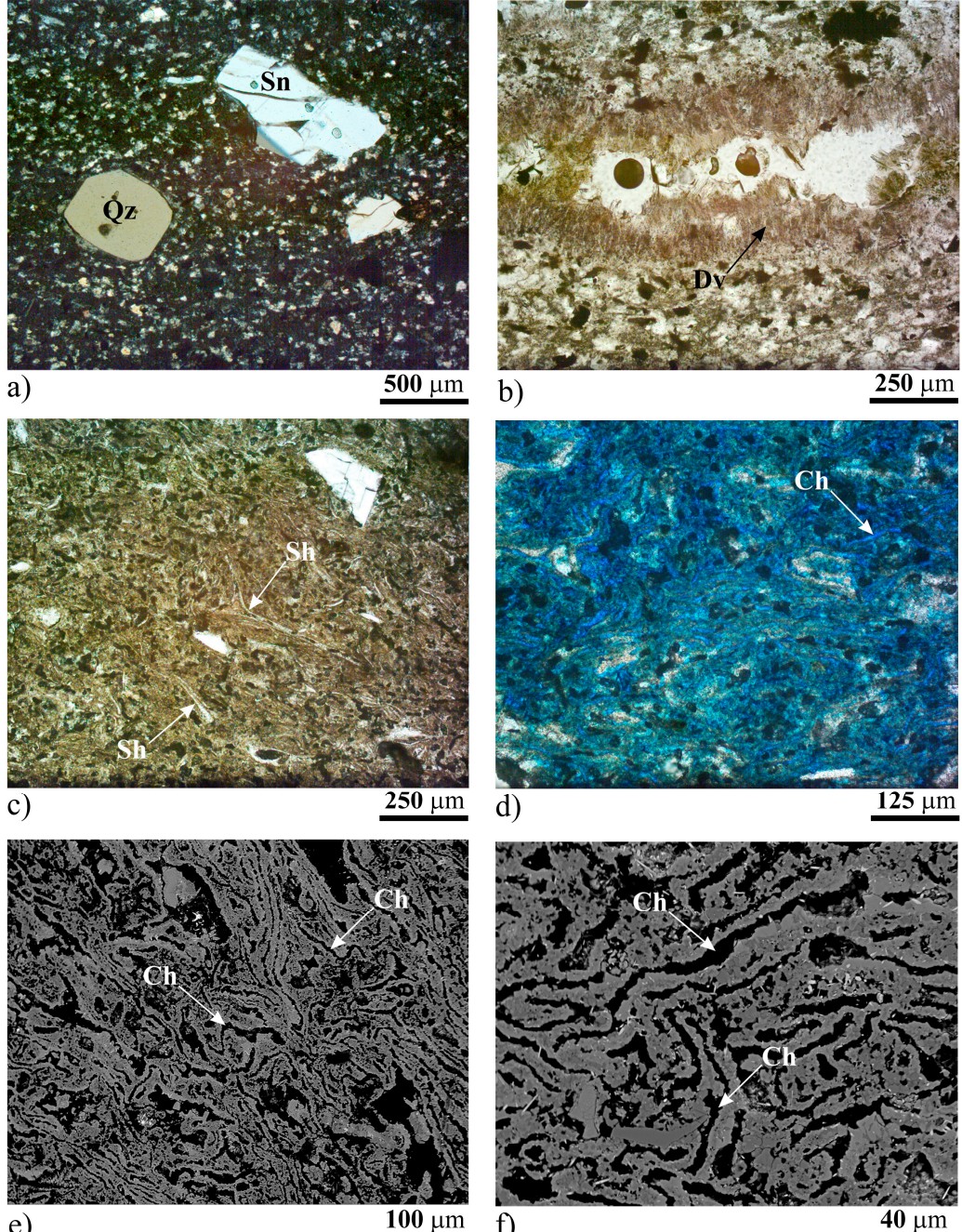

**Figure 6.** OM and SEM images of comendite lithotype from Sulcis area (SW Sardinia, Italy). (**a**) Sanidine (Sn) and quartz (Qz) phenocrysts, OM cross polarized light. (**b**) Devitrification (Dv) in the groundmass, OM plane polarized light. (**c**) Glass shards (Sh) in the groundmass, OM plane polarized light. (**d**) Channel porosity (Ch), OM plane polarized light, thin section treated with blue dye epoxy resin. (**e**) Channel porosity (Ch), SEM image. (**f**) channel porosity (Ch) in detail, SEM image.

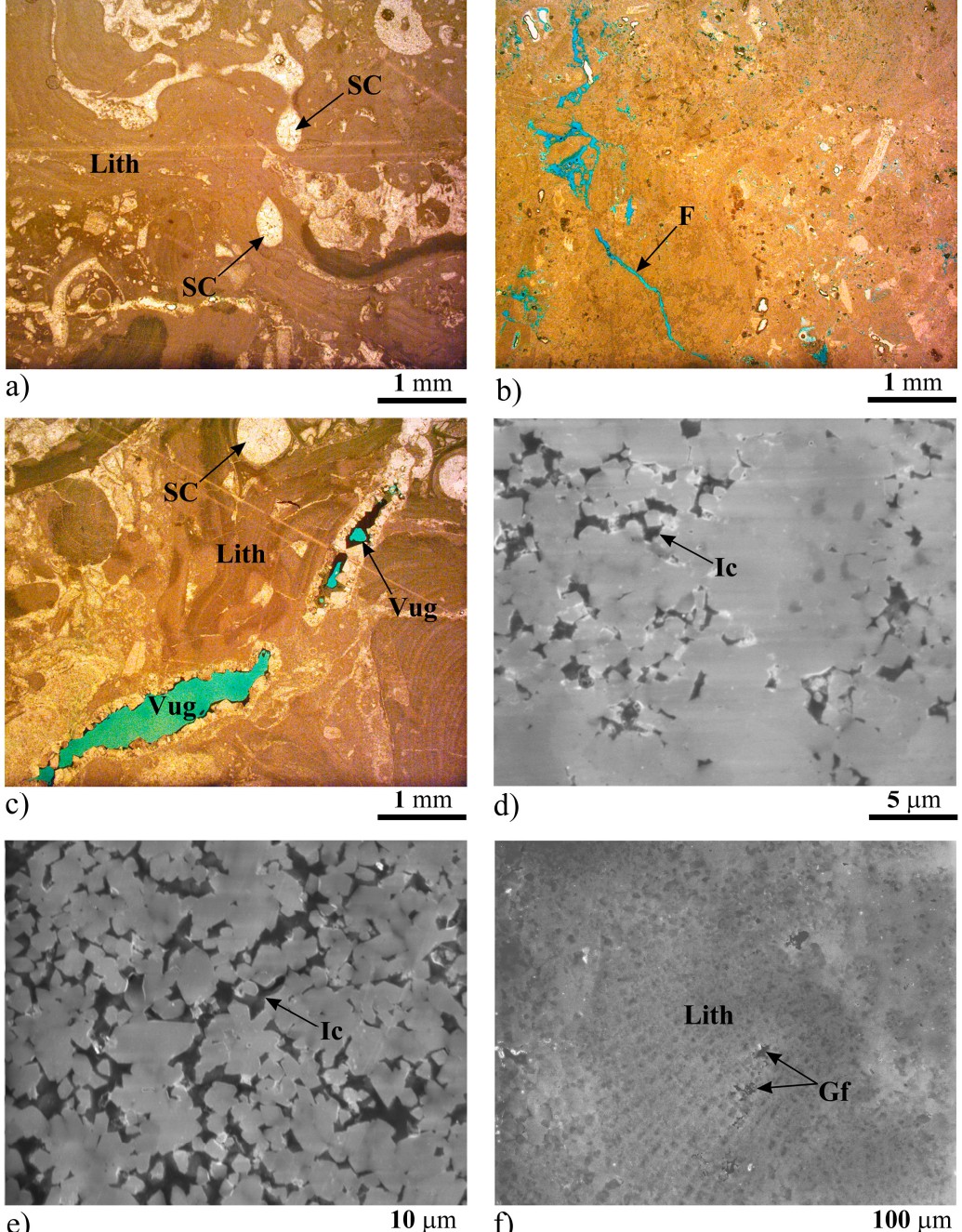

**Figure 7.** OM and SEM images of Pietra Forte shelf limestone from the city of Cagliari (S Sardinia, Italy). (**a**) Lithotamnium algae (Lith) and sparry calcite (SC), OM plane polarized light. (**b**) Fracture porosity (F), OM plane polarized light, thin section treated with blue dye epoxy resin. (**c**) Vug porosity (Vug) due to the dissolution phenomena, OM plane polarized light, thin section treated with blue dye epoxy resin. Lith (Lithotamnium algae), SC (sparry calcite). (**d**) Intercrystal porosity (Ic), SEM image. (**e**) Sector of the rock rich in intercrystal porosity (Ic), SEM image. (**f**) Growth framework (Gf) porosity within a Lithotamnium (Lith), SEM image.

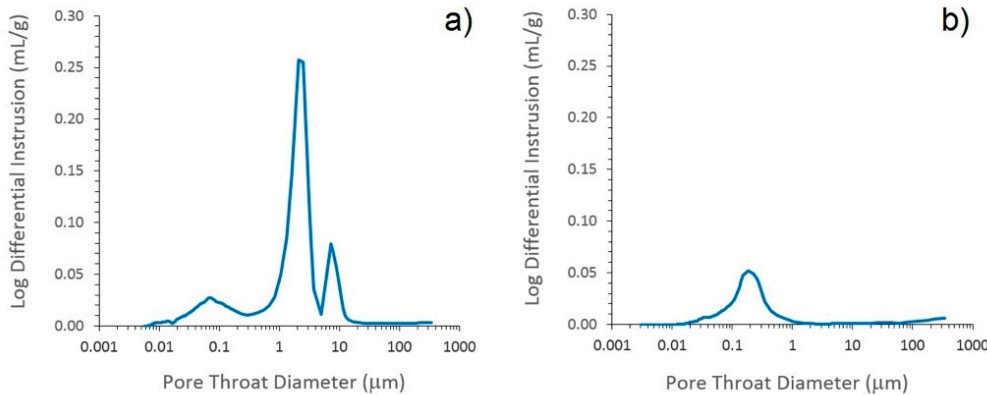

**Figure 8.** MIP curves highlighting the pore-throat size distributions in the analysed samples. (**a**) Pyroclastic comendite. (**b**) Pietra Forte limestone.

**Table 1.** MIP parameters of the studied samples.

| Sample | Median Pore-Throat Diameter (μm) | Bulk Density (g/cm³) | Skeletal Density (g/cm³) | Porosity (%) | Permeability (mD) | Tortuosity |
|---|---|---|---|---|---|---|
| Comendite | 2.09 | 1.74 | 2.30 | 24.61 | 1.53 | 36.22 |
| Pietra Forte | 0.20 | 2.46 | 2.71 | 9.24 | 0.01 | 6.55 |

The pore-throat size distribution for the Pietra Forte limestone is highlighted by the MIP curve of Figure 8b that shows the prevalence of pore-throats in the range 0.15 μm to 0.40 μm. The median pore-throat size (Table 1) is 0.20 μm. The tortuosity value (6.55, Table 1) can be related to a not very articulated pore network. The values of bulk and skeletal densities are 2.46 g/cm³ and 2.71 g/cm³, respectively (Table 1). The effective porosity is 9.24% and permeability is 0.01 mD (Table 1). The low permeability is mainly related to the presence of a compact carbonate cement as observed in OM and SEM analyses. This kind of cement hinders the passage of water within the rock.

*3.2. Geophysical Analyses*

The interpretation of the integrated geophysical data (Figures 9a–g and 10a–g) proposed in this work needs to be constrained by petrographical and petrophysical data. The textural and compositional characteristics of the rocks, as well as the type, quantity and size of the pores affect the geophysical responses, as already recognized in previous works [11,45,69]. Therefore, the geophysical interpretation needs to be integrated with petrographical data in order to maximise the benefit that derives from applying the proposed non-destructive multi-technique approach for the characterization of stone materials. In practical applications the recognition of physical property anomalies of the materials in a non-invasive way is of paramount importance and a good knowledge of the petrographical characteristics can help understand the origin of such anomalies and relate them to the intrinsic properties of the materials. The composition, texture and porosity predispose a stone material to different types of degradation and defects, which can be qualified and quantified both superficially and in depth using the geophysical methods applied in the present study. As a general remark we need to point out that the patterns of the geometrical anomalies computed with the help of the two methodologies TLS and SfM photogrammetry on the studied samples show interesting analogies. In fact, in spite of the fact that resolution of the two methods is different, the general shape of the residuals is quantitatively comparable [11].

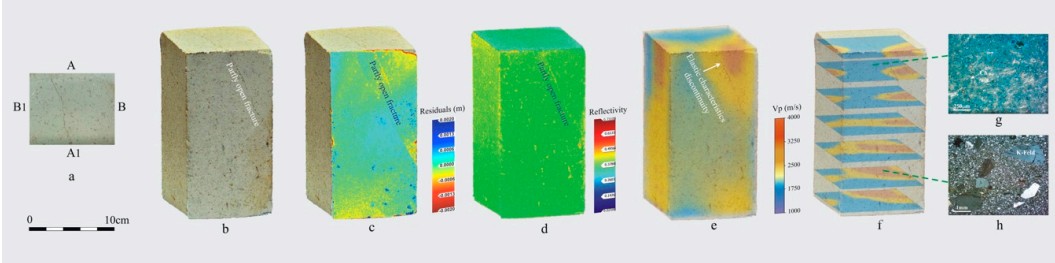

**Figure 9.** Results of 3D analyses by the multi-technique approach in the Comendite sample.
(**a**) Localization of the different faces of the sample. (**b**) 3D photogrammetric model. (**c**) Fit plane and
residuals. (**d**) 3D reflectivity map. (**e**) 3D ultrasonic tomography. (**f**) 3D ultrasonic tomography slices.
(**g**) Microscopic features of a porous sector of the sample, OM plane polarized light, thin section treated
with blue dye epoxy resin. Ch (channel porosity). (**h**) Microscopic features of a compact sector of the
sample, OM cross polarized light. Sa (sanidine).

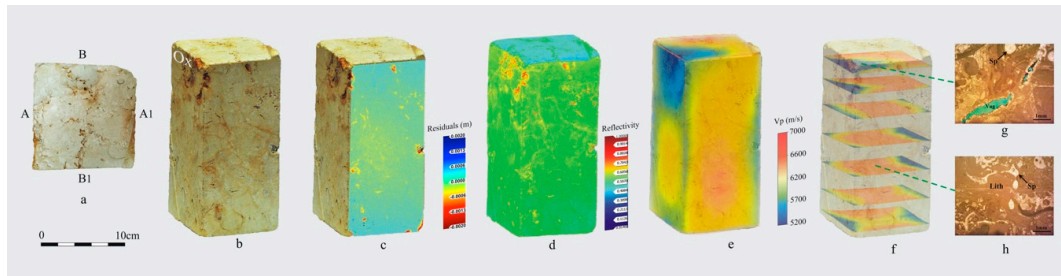

**Figure 10.** Results of 3D analyses by the multi-technique approach in the Pietra Forte sample.
(**a**) Localization of the different faces of the sample. (**b**) 3D photogrammetric model; Ox (oxidized
and macroporous zone). (**c**) Fit plane and residuals. (**d**) 3D reflectivity map. (**e**) 3D ultrasonic
tomography. (**f**) 3D ultrasonic tomography slices. (**g**) Microscopic features of a porous sector of the
sample, OM plane polarized light, thin section treated with blue dye epoxy resin. Vug (vug porosity),
Sp (sparry calcite). (**h**) Microscopic features of a compact sector of the sample, OM cross polarized light.
Lith (Lithotamnium algae), Sp (sparry calcite).

In particular, the geometrical anomalies and the reflectivity maps of the analysed samples
(Figure 9c,d and Figure 10c,d) are influenced by the different composition, textural characteristics
and porosity of the two lithotypes, as well as by the presence of discontinuities of a different nature.
In the comendite sample the higher negative geometrical anomalies (−0.5 mm to −2 mm) and higher
reflectivity of up to 0.73 mainly correspond to a fracture that develops in the central-upper part of
the sample. Variations in fracture surface roughness and aperture result in variations in geometrical
anomalies and reflectivity values (Figure 9c,d). From the analysis of the above maps it can be deduced
that only small parts of the fracture turn out to be open towards the top of the sample. In fact, the open
parts of the fracture are characterized by the higher negative geometrical anomalies evidenced by
the yellow/red colour and by a higher reflectivity. Geometrical anomalies of around 0 mm and lower
reflectivity values (about 0.4) detected on many other sectors of the sample correspond to almost
homogeneous materials without relevant defects or discontinuities. Especially reflectivity appears to
be affected by the uniformity of the textural and petrophysical characteristics of this rock.

In the Pietra Forte carbonate sample, the negative geometrical anomalies (about −1 mm) and the
higher reflectivity (range 0.6–1.0) that can be seen in the upper left sector of the sample (Figure 10c,d)
correspond to the presence of macropores that are due to dissolution phenomena of the calcite
and to an oxidation zone (Figure 10b). In the central and lower part of the sample (Figures 4
and 10c,d) the geometrical anomalies and reflectivity are lower (close to 0 mm and lower than 0.5,
respectively) and uniformly distributed due to the compactness and the high degree of cementation
that reduce the porosity and discontinuities on this material.

The geometrical anomalies and the reflectivity maps were also observed in the light of the results of the 3D ultrasonic tomography (Figures 9e and 10e). A few tomographic slices through corresponding horizontal sections of the investigated samples are shown in Figures 9f and 10f to facilitate the diagnostic process. The location and orientation of the horizontal slices (Figures 9f and 10f) extracted from the 3D ultrasonic tomography data volume were decided interactively for a clearer visualization of the internal distribution of the elastic characteristics of the investigated samples, especially considering the heterogeneity of the materials and the vertical development of the main fissures or discontinuities as can be seen both from the SfM photogrammetric and laser scanner analyses on the superficial parts of the samples.

Looking at the results of the 3D ultrasonic tomography on the comendite sample, the longitudinal ultrasonic velocity values are mainly related to the petrophysical characteristics of this rock and to the groundmass texture [94]. The lower velocity zones are characterized by a groundmass with an eutaxitic texture rich in pores of the channel type (Figure 9g), as observed by optical and electronic microscopy.

In these zones the widespread porosity causes structural attenuation phenomena of the ultrasonic signals and a decrease in the longitudinal velocity values (blue zones in Figure 9e,f). The high longitudinal velocity values (orange zones in Figure 9f) are related to a lower porosity and presence of compact groundmass made up of feldspars microcrystals closely tight to the vitreous fraction (Figure 9h). The increase in compactness in the central-lower part of the sample (Figure 9f) due to a more compact and poorly porous groundmass improves the elastic-mechanical characteristics of this rock and favours the propagation of the ultrasonic signals with increasing longitudinal velocity (orange zones, Figure 9e,f).

Furthermore, the results of the 3D ultrasonic tomography indicate that the shallow partly open fracture detected by the geometrical anomalies and TLS reflectivity corresponds in depth to a discontinuity of elastic characteristics. This discontinuity developing towards the upper part of the sample, bounds a 3D zone of low velocity (blue zones, Figure 9f) probably related to a fracture and altered zone where porosity increases.

In the Pietra Forte limestone the attenuation phenomena of the ultrasonic signals mostly occur in the upper part of the sample where the rock is oxidized and affected by macropores (Figure 10b, zone Ox). Therefore, in this zone a decrease in the longitudinal ultrasonic velocity occurs (blue zones in Figure 10e,f). The same zone is characterized in the surface by negative geometrical anomalies (Figure 10c) and high reflectivity values (Figure 10d). Considering the thin section analysis it can be deduced that this zone is mainly characterized by a secondary porosity of the vug type (Figure 10g), related to the dissolution of the calcite. The evolution of the dissolution process causes an increase in size and an interconnection of these pores, giving rise to a karst system that characterizes this lithology at the macroscale. The zones where crystallization of sparry calcite prevail (Figure 10h) are very compact and have good elastic characteristics, as indicated by longitudinal ultrasonic velocity values higher than 6500 m/s (orange-red zones in Figure 10e,f). These zones are characterized by a more uniform distribution of geometrical anomalies and reflectivity values on their surface.

## 4. Conclusions

The non-invasive approach for the characterization of different rock samples by means of different techniques can efficiently highlight the characteristics of the stone materials. The radiometric information available, such as TLS reflectivity maps, SfM photogrammetry and TLS high-resolution 3D models and patterns of geometrical residuals, were interpreted together with a comparison of reflectance and natural color anomalies on the surface materials in order to underline material anomalies. In fact, it was found that both geometrical and reflectivity anomalies are linked to the presence of textural heterogeneity, porosity variation, fissures, discontinuities or defects of a different nature. The faithful scale reproduction of the analysed samples and the high precision in the position and spatial distribution of the different geometric anomalies and reflectivity values also represented an extremely important data set for the best design and acquisition of the 3D ultrasonic tomography.

The latter was effective in providing an accurate image of the longitudinal velocity distribution inside the samples. The 3D ultrasonic tomographic model can be viewed and interpreted interactively from many perspectives enhancing the analysis and allowing the fine characterization of the sample materials in terms of elastic properties and the determination of the precise location and size of defects or damaged zones. Furthermore, the results of the 3D ultrasonic tomography highlighted the close connection between defects (e.g., fractures and macropores) in the surface materials and the worsening of the elastic characteristics in the corresponding inner parts of the samples.

The integration of the above three geophysical non-invasive techniques correlated with petrographical and petrophysical data represents a powerful high-resolution method for identifying the heterogeneity of the rocks, quantifying the amount of damages or defects at a different scale and calibrating in situ measurements.

The proposed geophysical non-invasive approach can give a useful contribution in the interpretation of geophysical data in well logging, core analysis, mining and environmental geophysics, geoengineering and related disciplines where the knowledge of the distribution of the elastic properties within a rock is one of the fundamental parameters. In the proposed procedure the combination of the shallow materials properties such as TLS reflectance or geometrical anomalies with the elastic properties of the inner materials can be incorporated into practical applications such as cultural heritage or quarrying activities. For instance, in the quarrying activities the propagation velocity of the longitudinal wave within a material is a good indicator of a rock drillability for a given rock type, and there is a good correlation between longitudinal wave velocity and penetration rates for diamond and percussion drilling.

Furthermore, the proposed non-invasive approach allows scientific reproducibility of the measurements at the same laboratory conditions and gives valuable information about potential changes in time in the rock condition.

**Author Contributions:** S.F. and G.C. conceived the new integrated methodology; F.C. carried out the photogrammetric survey and performed OM and SEM analyses; S.F. processed and interpreted the ultrasonic data and with F.C. performed the integrated ultrasonic and petrographic analysis; P.L. interpreted the mercury porosimetry data; G.C. and M.G.B. processed and interpreted TLS and photogrammetric data. All authors analysed and discussed the results, contributed to drafting the manuscript and preparing the figures. All authors reviewed and approved the manuscript.

**Funding:** Consorzio Interuniversitario per l'Ingegneria delle Georisorse (CINIGeo), Roma, Italy—research contract n. 264-2017 and RAS/FBS (grant number: F71/17000190002) grants for funding.

**Conflicts of Interest:** The authors declare no conflict of interest. The funders had no role in the design of the study; in the collection, analyses, or interpretation of data; in the writing of the manuscript, or in the decision to publish the results.

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
