# Peer review of "Characterization of Rock Samples by A High-Resolution Multi-Technique Non-Invasive Approach"

_minerals, doi:10.3390/min9110664_

Round 1

Reviewer 1 Report

This research is very interesting and well presented, the methodological approach scientifically accurate and thoroughly planned, and the interpretation of the results convincing and complete.

However, I recommend the authors to revise the presentation of the introduction and scientific background, and add few technical details about the proposed methodology and considerations about its applicability. I also commented on the discussion about the contribution of porosity. Details follow.

Introduction

This is presented as a methodological article, so it is important to stress the background and what is the novelty. I think that the literature discussion is not sufficient. I would start discussing briefly the methods of non-destructive rock characterization adopted in previous researches, mentioning the most used and successful techniques. If the authors prefer so, they might even only focus on the application of imaging techniques (not only those used here, but also, just to name few, thermography, X-ray mapping by µ-XRF, µ-CT, etc.). The three main techniques used in this research, in particular, should have a general technical description. A part of it might be just cut-pasted from the Methods section. Also, why did they were chosen over others? Finally, in the last part of the Introduction, I would place the aim of this research and its novelty. As it is now, the Introduction mostly contains information that should be rather added in the description of the materials (2nd paragraph indentation) and methods (3rd indentation). That information should be left in the introduction only much more synthetized, where the authors would describe research and aim.

Methods

1) The way the authors describe how they obtained the combined SfM-TLS maps is clear and rich of details. However, I would add some basic definitions in the text. About the geometric anomalies, when writing about the statistical procedure of inspection, I would add the definition of residual, together with the range of values and unit of measurement used here, which then you can read in Fig. 4. Similarly, I would also add the definition of reflectivity.

2) Figure 4: please make clear which faces of the samples are showed. I can see that, for the comendite, it is probably the face A1, and for Pietra Forte it is the right face in Fig. 3b-2. Please make this explicit or, even better, add the non-mapped natural-color photograph of the sample face next to that of the mapped faces (something like Fig. 9 and 10). In this way, one can quickly make a confrontation.

3) How the authors obtained a “3D representation of the distribution of the longitudinal wave velocity inside the investigated samples” via software, starting from the discrete measurements of velocity, is not entirely clear to me.

Discussion

1) Please make explicit what you think is the effect of the porosity determined by MIP on the maps. The maximum pore size detectable by the instrumentation used by the authors is 360 µm, which is a dimension too low for matching the resolution power of the techniques used I guess. Even though MIP is still important as a routine technique of basic petrophysical characterization, the relevant information in this case do not appear very useful for interpreting the reflectivity and residual maps. Not only concerning the differences in pore-size distribution, but also the values of effective porosity (although different concentrations of small pores might be theoretically viewed as “bulk” differences in the maps of the different rock types). The comendite is much more porous, but the areas more “sound” of the sample surfaces have practically the same appearance in the maps as Pietra Forte, with only minor differences. On the other hand, the pore-size range not actually investigated by MIP, and especially that of the larger macropores, gives a much stronger contribution to the parameter variations in the maps. And for this size class, macroscopic observations and/or thin-section observations, would be enough.

2) Lines 345-349: is it really correct talking about variations of the welding degree within the same sample of laboratory size? Sure, higher velocities relate to higher density, but this might be caused, excluding the contribution of discontinuities (their absence), by local minor variations of the porphyritic index, grain size of phenocrysts, groundmass texture, etc.

Conclusions

The methodology proposed is demanding in terms of equipment costs (at least a part of it), and looks so even in terms of time for data processing (some relevant brief information about how much automated the procedure is, added in Methods, would be interesting). What kinds of application might justify these efforts? The authors should discuss this point.

At the scale of laboratory specimens and sample characterization, the information obtained from the residual and reflectivity maps, which is limited to the surface (or very near surface) is almost the same of what one may find out after a careful but simple macroscopic observation. The information obtained with the ultrasonic tomography, instead, is related to “invisible” rock layers and, therefore, very useful: but this is already a well-established laboratory and field technique!

Therefore, I guess the most useful and fruitful application would be on rock masses or, in the case of cultural heritage, large buildings made of a great number of different stone elements, aiding a global structural survey. Is this what the authors have in mind? If not, what do they suggest for a smart application of the techniques to real problems?

Author Response

Detailed Response to Reviewer1

We thank reviewer 1 for his positive comments on our work and answer his questions:

INTRODUCTION

Comment from the reviewer:

This is presented as a methodological article, so it is important to stress the background and what is the novelty. I think that the literature discussion is not sufficient. I would start discussing briefly the methods of non-destructive rock characterization adopted in previous researches, mentioning the most used and successful techniques. If the authors prefer so, they might even only focus on the application of imaging techniques (not only those used here, but also, just to name few, thermography, X-ray mapping by µ-XRF, µ-CT, etc.). The three main techniques used in this research, in particular, should have a general technical description. A part of it might be just cut-pasted from the Methods section. Also, why did they were chosen over others? Finally, in the last part of the Introduction, I would place the aim of this research and its novelty. As it is now, the Introduction mostly contains information that should be rather added in the description of the materials (2ndparagraph indentation) and methods (3rdindentation). That information should be left in the introduction only much more synthetized, where the authors would describe research and aim.

Authors’ response:

Following the suggestions of reviewer 1, we stressed the background on some non-destructive methods for rock characterization, mentioning mainly the imaging techniques application (inserted rows from 37 to 56 - revised version).

We also added a short description of the three non invasive techniques used in the research (see rows 71 to 85 and rows 94 to 98) as requested by reviewer 1.

In the introduction we shortly described the aim (see rows 61-63 - revised version) and we specified the novelty (rows 63-64 - revised version).

As requested by the reviewer, we synthesized the information of the investigated materials (deleted row 40-41 of the original version of the text), which are described in paragraph 2.1 sample materials. We also eliminated from the Introduction some sample details (size ect.): deleted rows 42-46 of the original text and moved them to the mentioned 2.1 paragraph of the revised text (see rows 120-124 - revised version).

METHODS

Comment from the reviewer:

1) The way the authors describe how they obtained the combined SfM-TLS maps is clear and rich of details. However, I would add some basic definitions in the text. About the geometric anomalies, when writing about the statistical procedure of inspection, I would add the definition of residual, together with the range of values and unit of measurement used here, which then you can read in Fig. 4. Similarly, I would also add the definition of reflectivity.

Authors' response:

Every request by reviewer 1 described in the above point 1 has been met (See rows.196-201 and 240-273 of the revised text)

Comment from the reviewer:

2) Figure 4: please make clear which faces of the samples are showed. I can see that, for the comendite, it is probably the face A1, and for Pietra Forte it is the right face in Fig. 3b-2. Please make this explicit or, even better, add the non-mapped natural-color photograph of the sample face next to that of the mapped faces (something like Fig. 9 and 10). In this way, one can quickly make a confrontation.

Authors' response:

The request of the reviewer was met: we added the non mapped natural color photograph of the sample face next to that of the mapped face (See the new Fig. 4) INGV.

Comment from the reviewer:

3) How the authors obtained a “3D representation of the distribution of the longitudinal wave velocity inside the investigated samples” via software, starting from the discrete measurements of velocity, is not entirely clear to me.

Authors' response:

To satisfy the reviewer's request, we have upgraded paragraph 2.2.4 inserting further details (see rows 313-328) on the procedure used for the calculation of the tomographic models.

DISCUSSION

Comment from the reviewer:

1) Please make explicit what you think is the effect of the porosity determined by MIP on the maps. The maximum pore size detectable by the instrumentation used by the authors is 360 µm, which is a dimension too low for matching the resolution power of the techniques used I guess. Even though MIP is still important as a routine technique of basic petrophysical characterization, the relevant information in this case do not appear very useful for interpreting the reflectivity and residual maps. Not only concerning the differences in pore-size distribution, but also the values of effective porosity (although different concentrations of small pores might be theoretically viewed as “bulk” differences in the maps of the different rock types). The comendite is much more porous, but the areas more “sound” of the sample surfaces have practically the same appearance in the maps as Pietra Forte, with only minor differences. On the other hand, the pore-size range not actually investigated by MIP, and especially that of the larger macropores, gives a much stronger contribution to the parameter variations in the maps. And for this size class, macroscopic observations and/or thin-section observations, would be enough.

Authors' response:

1a) The authors thank the reviewer for his interesting considerations on the usefulness of MIP analysis on the interpretation of reflectivity and residual maps. The Authors are in quite good agreement with the reviewer: it is difficult to find an easy and direct connection between the MIP results and both reflectivity and residual maps. However it should be considered that also the conditions of the surface materials are a consequence of the intrinsic characteristics of the rocks, such as microporosity and especially the distribution and amount of pore-throats. These latter represent the pathways of fluid circulation into the rock and determine permeability at the microscale. Furthermore, the authors’ idea was to use the MIP data to recover further information on the tortuosity and permeability. For instance a large tortuosity characterizes a medium where pore connectivity is poor and this characteristic influences the porosity and consequently the propagation of the acoustic signal. In fact porosity at all scales (from micro to macro) is one of the parameters of paramount importance in analyzing acoustic signal propagation, as also widely reported in the international literature (for instance: Anselmetti and Eberli, 1993; Kenter et al., 1995; Kenter et al., 2002; Weger et al., 2004; Verver et al., 2008; Brigaud et al., 2010; Torok and Vasarhelyi, 2010; and others).

1b) As specified in the text (see rows 91-92), the geometrical anomalies and reflectivity maps give objective information on the surface materials pointing out the presence of different kinds of materials anomalies. Macroscopic observations (visual examination) strictly depend on the personal expertise of the analyst and require robust training and experience. Consequently the results of the visual inspection can lose some objectivity and be difficult to communicate. These aspects could hinder scientific reproducibility and objectivity. Moreover, the macroscopic observations can be still valid as a tool for the preliminary analysis of surface materials.

Furthermore, as specified in the text (see rows 59-61 – paragraph INTRODUCTION), the above maps have been very useful in giving an adequate geometric support to rendering the 3D ultrasonic data at their precise location.

Comment from the reviewer:

2) Lines 345-349: is it really correct talking about variations of the welding degree within the same sample of laboratory size? Sure, higher velocities relate to higher density, but this might be caused, excluding the contribution of discontinuities (their absence), by local minor variations of the porphyritic index, grain size of phenocrysts, groundmass texture, etc.

Authors’ response:

The Authors thank the reviewer for his legitimate question. Following the reviewer's suggestion, the authors deleted the rows 345 and 349 of the original text and replaced them with a few new sentences (see rows 438, 451-452 and 455-456 of the revised version).

CONCLUSIONS

Comment from the reviewer:

1) The methodology proposed is demanding in terms of equipment costs (at least a part of it), and looks so even in terms of time for data processing (some relevant brief information about how much automated the procedure is, added in Methods, would be interesting). What kinds of application might justify these efforts? The authors should discuss this point.

Authors' response:

The proposed methodology is quite easy to apply and cost effective. We have added some information about this in the paragraph METHODS (rows72-78).

Many applications of the computer vision are present in literature, at any scale of dimensions, also regarding rock and minerals (see for example Snavely et alii; Westoby et alii; Ighlaut et alii, and references therein), mechanical parts of cars, boats, electronic devices, trees, and outcrops of geological strata. The resulting high resolution 3D models represent a valuable tool for technicians and experts to manage objects of complex form at their homes on their own computers. Scientist of every discipline can take their studied objects and send them anywhere on the internet. Visual inspections, precise measurements, movies and other interactive operations are possible if a good high resolution 3D model of an object under study is available. In our case, we studied two samples of 22-24, 9-12, 9-12 cm3 volumes weighing some Kilos not so easy to handle, but with the aid of 3D models any type of inspection is possible. Finally, the memory of shapes can be preserved for the future even, for example, after the studied objects have been destroyed.

A few sentences indicating the possible application and usefulness (elastic dynamic characterization of materials) are also reported in the INTRODUCTION (rows 59-63). Furthermore, following the reviewer’s request, the authors have inserted a few sentences to comment on the usefulness and fields of application of the proposed integrated methodology (rows 509 and 521 - revised version) in the paragraph “conclusions”.

Comment from the reviewer:

2) At the scale of laboratory specimens and sample characterization, the information obtained from the residual and reflectivity maps, which is limited to the surface (or very near surface) is almost the same of what one may find out after a careful but simple macroscopic observation. The information obtained with the ultrasonic tomography, instead, is related to “invisible” rock layers and, therefore, very useful: but this is already a well-established laboratory and field technique.

Authors’ response:

While acknowledging that macroscopic analysis (visual inspection) can be useful and performed in a very preliminary analysis of the surface materials, the authors do not agree with the reviewer as to the fact that this analysis can replace the high resolution 3D models obtained with the TLS and SfM techniques. This argument has already been discussed in previous replies given by the authors to the reviewer (see point 1b - DISCUSSION and point 1 - CONCLUSION of this document). A sentence regarding this aspect has been also inserted in the revised text (see rows 91-92 - paragraph INTRODUCTION of the revised text).

The authors do not understand the meaning of the second part of the comment of the reviewer, namely "the ultrasonic tomography ......... but this is already a well-established laboratory and field technique". Ultrasonic tomography has been applied for many years in different sectors of the applied sciences. This does not subtract any of the method’s validity. Indeed in our opinion it increases its potential. Many papers have been published in time on 2D and more recently 3D acoustic tomography to solve a number of problems in many research fields and the application of this technique is constantly used more widely. Moreover, techniques such as IR thermography, X-ray mapping by µ-XRF, µ-CT, etc., as mentioned by the reviewer in his first comment (INTRODUCTION), have been also applied for many years and are well established laboratory and field techniques. These techniques are still applied successfully today.

Reviewer 2 Report

In the introduction: the introduction lacks a careful bibliography on how the materials are characterized and on examples of works that use techniques similar to those proposed by the authors

From row 37 to 39 to 65 Explain the concept more broadly

From row 37 it is very risky to write in detail, when in detail one can learn about a material only after careful physical-chemical laboratory analysis, change the sentence

From row 40 to 65, this part of the text should be put into the methods, an introduction with a robust bibliographic treatment is missing

From row 55 to 6e insert bibliographic citation

From row 82, only modern construction or also historic buildings?, specify

Row 230 25% is total porosity? specify

From row 233, 2% is total porosity? Why porosity with MIP is higher?

In Conclusion, xplain how you can apply these methods in situ, I didn't understand it

The work is valid at an experimental level and with laboratory samples, but are these new and non-invasive methods really applicable on site for the characterization and verification of the state of conservation of the materials?

The 3d modeling is interesting when we want to understand in situ the Material loss due to decay or pesence of deposits, but on this samples  cannot see the usefulness for the characterization. Explain better the purposes in the paragraph  “ Terrestrial Laser Scanner and SfM Photogrammetry” And results

Author Response

Detailed Response to Reviewer2

We thank Reviewer2 for his positive comments on our work and answer his questions:

INTRODUCTION

1 - Comment from the reviewer:

- In the introduction: the introduction lacks a careful bibliography on how the materials are characterized and on examples of works that use techniques similar to those proposed by the authors

Authors’ response:

Following the reviewer’s comment the Authors have added bibliography on the rock samples characterization by non invasive techniques (see rows 46-52 of the revised text). The non invasive approach proposed by the Authors is new in the geophysical characterization of rock samples so it is difficult to add references referring to similar approaches as requested by reviewer 2. References on similar approaches for in situ inspection have been added (see row 66 of the revised text).

2 - Comment from the reviewer:

- From row 37 to 39 to 65 Explain the concept more broadly

Authors’ response:

Following the reviewer’s comment, in the paragraph INTRODUCTION we have upgraded the original text (see rows 37-56, 59-64, 71-85 and 94-98 of the revised text).

3 - Comment from the reviewer:

From row 37 it is very risky to write in detail, when in detail one can learn about a material only after careful physical-chemical laboratory analysis, change the sentence.

Authors' response:

The Authors agree with the reviewer. The rock characterization can be done with different methods and laboratory analyses mainly of chemical-physical types. The approach proposed in our work is that of a geophysical characterization and therefore included in the physical characterization. To avoid misunderstundings we have changed the sentence of row 37 of the original version of the text and added the adjective geophysical to characterization (row 58 of the revised text). In this way we specify that our approach is aimed at a geophysical characterization.

4- Comment from the reviewer:

From row 40 to 65, this part of the text should be put into the methods, an introduction with a robust

bibliographic treatment is missing.

Authors’ response:

Following the reviewers' suggestion, part of the text of the paragraph introduction (rows 40-46 of the original text) have been moved to the paragraph methods (rows 120-124 of the revised version). Furthermore many references have been inserted in the paragraph INTRODUCTION and in other paragraphs of the revised text.

5- Comment from the reviewer:

From row 55 to 6e insert bibliographic citation

Authors’ response:

The Authors have inserted further references as requested by the reviewer (see INTRODUCTION paragraph of the revised text).

6- Comment from the reviewer:

From row 82, only modern construction or also historic buildings?, specify

Authors’ response:

Complying with the reviewers' request, the Authors have specified that they are referring to historic buildings (see row 129 of the revised text).

7 - Comment from the reviewer:

Row 230 25% is total porosity? specify

Authors’ response:

The porosity (25%) is the porosity observed by OM and SEM analyses, therefore it is the total visible porosity not the total one (s.s). Consequently to conform to the reviewer’s request, we have substituted the term porosity with “visible porosity” (see row 335 and 339 of the revised text).

8 - Comment from the reviewer:

From row 233, 2% is total porosity? Why porosity with MIP is higher?

Authors’ response:

Regarding the Pietra Forte sample, the porosity value of 2% is not referred to total porosity because, as specified in the above row 335, it is referred to the visible porosity.

The difference (of about 7%) between the porosity values observed by the microscopy observations (OM, SEM) and the MIP analyses depends on the fact that the MIP technique measures pore-throat sizes that are hardly quantifiable by OM and SEM analyses.

9 - Comment from the reviewer:

In Conclusion, xplain how you can apply these methods in situ, I didn't understand it

Authors' response:

The paper proposes an integrated non-invasive approach to the geophysical characterization of rock samples in laboratory. Consequently, the in situ application could not be described in this context. The authors believe that by "in situ application", the reviewer2 means a diagnostic analysis on monumental structures. If this is the case, the Authors would like to specify that only recently a very similar approach has been used successfully in the Cultural Heritage field (see for instance the papers by Bianchi et al., 2018 and Donadio et al., 2018). These works have been cited in the references of the paper in question. There are no integrated approaches of the same type in previous literature.

Round 2

Reviewer 1 Report

The authors have addressed in a careful and satisfactory way all of the comments I raised in the first round of review. Therefore I now recommend the publication of the paper in its present state.